# Mechanisms of Cell Death Induced by Erastin in Human Ovarian Tumor Cells

**DOI:** 10.3390/ijms25168666

**Published:** 2024-08-08

**Authors:** Birandra K. Sinha, Carri Murphy, Shalyn M. Brown, Brian B. Silver, Erik J. Tokar, Carl D. Bortner

**Affiliations:** 1Mechanistic Toxicology Branch, Division of Translational Toxicology, National Institutes of Environmental Health, NIH, Research Triangle Park, NC 27709, USA; carri.murphy@nih.gov (C.M.); shalyn.brown@nih.gov (S.M.B.); brian.silver@nih.gov (B.B.S.); erik.tokar@nih.gov (E.J.T.); 2Laboratory of Signal Transduction, National Institutes of Environmental Health, NIH, Research Triangle Park, NC 27709, USA; bortner@niehs.nih.gov

**Keywords:** erastin, ferroptosis, ovarian tumor cells, RSL3, reactive oxygen species

## Abstract

Erastin (ER) induces cell death through the formation of reactive oxygen species (ROS), resulting in ferroptosis. Ferroptosis is characterized by an accumulation of ROS within the cell, leading to an iron-dependent oxidative damage-mediated cell death. ER-induced ferroptosis may have potential as an alternative for ovarian cancers that have become resistant due to the presence of Ras mutation or multi-drug resistance1 (MDR1) gene expression. We used K-Ras mutant human ovarian tumor OVCAR-8 and NCI/ADR-RES, P-glycoprotein-expressing cells, to study the mechanisms of ER-induced cell death. We used these cell lines as NCI/ADR-RES cells also overexpresses superoxide dismutase, catalase, glutathione peroxidase, and transferase compared to OVCAR-8 cells, leading to the detoxification of reactive oxygen species. We found that ER was similarly cytotoxic to both cells. Ferrostatin, an inhibitor of ferroptosis, reduced ER cytotoxicity. In contrast, RSL3 (RAS-Selective Ligand3), an inducer of ferroptosis, markedly enhanced ER cytotoxicity in both cells. More ROS was detected in OVCAR-8 cells than NCI/ADR-RES cells, causing more malondialdehyde (MDA) formation in OVCAR-8 cells than in NCI/ADR-RES cells. RSL3, which was more cytotoxic to NCI/ADR-RES cells, significantly enhanced MDA formation in both cells, suggesting that glutathione peroxidase 4 (*GPX4*) was involved in ER-mediated ferroptosis. ER treatment modulated several ferroptosis-related genes (e.g., *CHAC1*, *GSR*, and *HMOX1/OX1*) in both cells. Our study indicates that ER-induced ferroptotic cell death may be mediated similarly in both NCI/ADR-RES and OVCAR-8 cells. Additionally, our results indicate that ER is not a substrate of P-gp and that combinations of ER and RSL3 may hold promise as more effective treatment routes for ovarian cancers, including those that are resistant to other current therapeutic agents.

## 1. Introduction

Ovarian cancer affects a significant number of women worldwide and is a leading cause of death among gynecological cancers. It is also considered one of the most lethal malignancies due to its high recurrence rate and inadequate early detection methods. Ovarian cancers have been reported to be difficult to treat due to ineffective screening strategies and delayed diagnosis [1]. Ovarian cancers are highly heterogeneous, with distinct etiology, morphology, molecular biology, and prognosis. Unfortunately, they are often treated as a single cancer [2,3]. Most patients receiving standard-of-care therapy (cytoreductive surgery followed by adjuvant chemotherapy) develop chemotherapy resistance, resulting in a poor survival rate of 30–40% worldwide. While the initial response rate to chemotherapy is high, most patients become resistance to chemotherapy due to the presence of multi-drug resistance P-170 glycoproteins that are highly expressed in various clinical tumor samples. In addition, Fantone et al. [4] have indicated that SLC7A11 (also known as xCT) expression is significantly upregulated in ovarian cancers, resulting in the inhibition of ferroptosis, enhancing cancer cell proliferation, invasion, and chemoresistance. Thus, the emergence of chemotherapy resistance and failure to respond to chemotherapy poses significant challenges in the treatment of ovarian cancers, requiring innovative approaches for therapy development for more effective anti-cancer agents or combinations of drugs.

Recently, a clinical trial led to the FDA granting accelerated approval of the first antibody drug conjugate (ADC), mirvetuximab soravtansine, for the treatment of platinum-resistant ovarian cancer. Known as the MIRASOL trial, this clinical trial evaluated mirvetuximab soravtansine, an ADC designed to target folate receptor alpha, for the treatment of platinum-resistant ovarian cancer. Patients receiving the ADC lived longer and experienced fewer and less serious side effects [5]. However, this treatment did not fully cure patients or enable long-term survival [5].

Recently, drugs that induce tumor cell death through ferroptosis have been suggested to represent a promising avenue for cancer therapy, especially in cancers that are resistant to conventional treatments [6,7]. Ferroptosis, a form of regulated cell death, is characterized by the accumulation of lipid peroxides and dependence on iron. The damaging species is reactive ^●^OH, formed from the reaction of H_2_O_2_ with Fe^2+^ (the Fenton reaction). Ferroptosis-based cell death has been shown to be different from other forms cellular death such as necrosis, autophagy, and apoptosis [8,9]. The cellular death resulting from ferroptosis arises from the inhibition of glutathione peroxidase 4 (GPX4) and the accumulation of intracellular lipid hydroperoxides, causing damage to cellular membranes in the presence of iron [10,11]. ER-induced ferroptosis of tumor cells primarily involves disrupting the cellular antioxidant defense system and triggering lipid peroxidation. Several mechanisms of ER-dependent ferroptosis have also been suggested, including the inhibition of xCT, the inhibition of the mitochondria-bound voltage-dependent anion channel (VDAC), and the modulation of the tumor suppressor p53 gene [7,12,13,14].

The xCT system transporter plays a crucial role in maintaining intracellular levels of cystine, a precursor for the synthesis of the antioxidant glutathione (GSH). Inhibition of the xCT system by ER leads to a decrease in intracellular cystine and, subsequently, a decrease in GSH levels. This decrease in cellular GSH results in an increase in oxidative stress due to ROS/RNS formation, causing cellular damage and death. In addition, ER has been reported to inhibit VDAC, which plays an important role in the induction of ferroptosis. VDAC, an ion channel located in the outer mitochondrial membrane, mediates and controls molecular and ion exchange between the mitochondria and the cytoplasm. The permeability of the VDAC can be pharmacologically altered, causing mitochondrial metabolic dysfunction, ROS production, oxidative damage, and cell death [15].

ER and its analogs have been shown to sensitize resistant tumor cells to chemotherapy agents [16,17]. Furthermore, ER has shown activity in ovarian cancers in vitro. Therefore, understanding the mechanisms of ER cytotoxicity is essential for developing strategies to design better drug combinations for the treatment of human cancers. Here, we have utilized both OVCAR-8 and OVCAR-8-derived Adriamycin (ADR)-resistant (NCI/ADR-RES) cell lines to understand and decipher the mechanisms of ER cytotoxicity. We used these ovarian cell lines as models to evaluate the mechanisms of ER cytotoxicity. First, NCI/ADR-RES cells are extremely resistant to free radical-generating drugs due to the presence of higher expressions of SOD and catalase, compared to OVCAR-8 cells [18,19]. Also, NCI/ADR-RES cells have higher expressions of both glutathione peroxidase (GPX1) and glutathione transferase (GST) compared to OVCAR-8 cells. These antioxidant enzymes and GSH/GST systems protect cells from oxidative damage resulting from the formation of ROS in tumor cells [20,21]. Therefore, we reasoned that if ER cytotoxicity is dependent upon ROS formation, then ER would be more cytotoxic to OVCAR-8 than NCI/ADR-RES cells, which more readily detoxify reactive species by glutathione peroxidase, SOD, and catalase. We reasoned that elucidating the mechanisms of ER-induced cell death will aid in the design of better therapeutic combinations. For example, ferroptosis inducers such ER or RSL3 could be combined with traditional chemotherapeutics for the treatment of resistant ovarian cancers in the clinic.

## 2. Results

### 2.1. Cytotoxicity Studies with Erastin

Previous studies have shown that ER is cytotoxic to human ovarian tumor cells [22]. In this study, ER was found to be highly cytotoxic to both human OVCAR-8 and its P-gp- expressing resistant NCI/ADR-RES cells (Figure 1). There were small (non-significant) differences in cytotoxicity, as found with the TiterGlo assay (IC_50_ 1.2 ± 0.10 × 10^−6^ M vs. 0.8 ± 0.15 × 10^−6^ M, for OVCAR-8 and NCI/ADR-RES cells, respectively). Intriguingly, we found resistant cells to be slightly more sensitive to ER. This is in contrast to previous findings using other P-gp-expressing ovarian tumor cells, in which Taxol-resistant ovarian cells were also resistant to ER [23].

### 2.2. Effects of Ferrostatin-1 and RSL3 on Erastin Cytotoxicity

Since ER is a known inducer of ferroptosis, we used both Ferrostatin-1 (FES), an inhibitor of ferroptosis [24,25,26], and RSL3, an inducer of ferroptosis [27,28], to evaluate ER cytotoxicity in OVCAR-8 or NCI/ADR-RES cells cell lines. As shown in Figure 2, FES significantly attenuated ER cytotoxicity, while RSL3 markedly enhanced ER cytotoxicity in both cells. This strongly suggests that ER mediates cellular death through ferroptosis in these cells. ER is also known to induce apoptosis in certain tumor cell types. However, neither the Annexin binding assay [29] nor CaspaTag assay [30] showed any apoptosis induced by ER in either OVCAR-8 or NCI/ADR-RES cells.

### 2.3. Cytotoxicity of RSL3 in OVCAR-8 and NIH/ADR-RES Cells

As RSL3 appeared to be differentially more cytotoxic to NIH/ADR-RES cells (Figure 2), we further investigated the cytotoxicity of RSL3 in these ovarian cells. We found that RSL3 is significantly more cytotoxic to the resistant NIH/ADR-RES cells than OVCAR-8 cells (Figure 3).

### 2.4. Effects of ER on Xc-Transporter

ER is known to inhibit the Xc-transporter in some tumor cells, leading to decreases in cellular glutathione and oxidative damage [7]. We, therefore, examined the effects of ER on the xCT-transporter in both OVCAR-8 and NCI/ADR-RES cells. The data depicted in Figure 4 clearly indicate that ER effectively inhibits the xCT transporter in both OVCAR-8 and NCI/ADR-RES tumor cells. Notably, as little as 1.0 µM ER was sufficient to inhibit the xCT-transporter in both cell lines.

### 2.5. ROS Formation by ER in OVCAR-8 and NIH/ADR-RES Cells

ER has been reported to generate ROS in some tumor cells, leading to oxidative damage and cell death. Mitosox was utilized to detect the formation of ER-dependent ROS species in both OVCAR-8 and NCI/ADR-RES cells. Significant amounts of ROS were detected in OVCAR-8 cells following ER treatment for 4 h. Little or no ROS could be detected in NCI/ADR-RES cells (Figure 5), most likely from a rapid detoxification of active species formed due to the presence of SOD, catalase, GPX1, and GST expressed in the NCI/ADR-RES cells. We have previously reported similar findings with NCX4040 in these cell lines [22].

### 2.6. ER Induces Lipid Peroxidation in OVCAR-8 Ovarian Cells

We examined the formation of lipid peroxides by ER in OVCAR-8 and NIH/ADR-RES cells. As shown in Figure 6, a small but significant amount of peroxidation, measured as MDA, was detected in OVCAR-8 cells compared to the resistant variant. Adriamycin, a known inducer of peroxidation, was included as a positive control. MDA formation was dose-dependent in OVCAR-8 cells. In contrast, neither ADR nor ER significantly induced the peroxidation of lipids in the resistant NCI/ADR-RES cells (Figure 6A).

### 2.7. RSL3 Enhances ER-Induced Lipid Peroxidation in OVCAR-8 and NIH/ADR-RES Cells

Because RSL3 enhanced ER cytotoxicity in these ovarian cells and we found RSL3 to be more cytotoxic to NCI/ADR-RES cells than OVCAR-8 cells, we investigated if RSL3 would also selectively enhance lipid peroxide formation in the resistant NCI/ADR-RES cells. Our results (Figure 6B) indicate that RSL3 enhanced lipid peroxidation in the NCI/ADR-RES cells such that there was no significant difference between the OVCAR-8 and NCI/ADR-RES cells in MDA formation. RSL3 also increased MDA formation in the OVCAR-8 cells (Figure 6B).

### 2.8. N-Acetyl Cysteine (NAC) Attenuates ER Cytotoxicity

Since ER-mediated cytotoxicity in tumor cells results from ROS formation, we evaluated the cytotoxicity of ER in the OVCAR-8 and NCI/ADR-RES cells in the presence of NAC, a known scavenger of ROS and ferroptosis [31,32]. Our results (Figure 7) show that NAC significantly attenuated the cytotoxicity of ER in both cell lines, suggesting that ER cytotoxicity was mediated by ROS generation in these cells.

### 2.9. Gene Expression Changes in Response to ER in OVCAR-8 and NCI/ADR-RES Cells

RT-PCR was utilized to examine the expression levels of genes related to oxidative damage, ferroptosis, and oxidative DNA damage repair, specifically 8-oxoguanine DNA glycosylase 1 (*OGG1*), following the treatment of OVCAR-8 and NCI/ADR-RES cells with ER (2.5 µM) for 4 h and 24 h. We found that *OGG1* was rapidly induced by ER in OVCAR-8 cells (4 h) and decreased at 24 h. *OGG1* was significantly induced only at 24 h in NCI/ADR-RES cells. These observations suggest that ER caused oxidative DNA damage in both cell lines. The expression of anti-apoptotic gene *BCl2* was unchanged in both the OVCAR-8 and NCI/ADR-RES cells, while the pro-apoptotic *BAX* gene expression was marginally decreased in both cells.

ER treatment also resulted in significant modulations of several genes related to ferroptosis and oxidative stress in both ovarian cell lines (Figure 8). Heme oxygenase (*HMOX1/OX1*), a biomarker for oxidative stress [33,34], was significantly induced by ER treatment in both OVCAR-8 and NCI/ADR-RES cells. In contrast, *NOX4*, a gene responsible for generating superoxide anion radicals (O_2_^.−^), was not affected in neither the OVCAR-8 or NCI/ADR-RES cells (Figure 8). *CHAC1*, a biomarker for ferroptosis [35,36,37], was highly induced in both OVCAR-8 and NCI/ADR-RES ovarian tumor cells. *GPX4*, a key ferroptosis indicator in cells and responsible for suppressing the process of ferroptosis [27,38], was decreased in the OVCAR-8 cells at 4 h and 24 h but was not significantly affected in the NCI/ADR-RES cells. Glutathione reductase (GSR), responsible for maintaining GSH hemostasis, was decreased only in the NCI/ADR-RES cells. Nuclear erythroid factor 2 (NRF2), the master regulator of antioxidant pathways [39,40], was unchanged in both cells. SLC7A11, which is responsible for GSH synthesis and oxidative stress in cells and is highly expressed in ovarian tumor cells [4,41], was unchanged in the OVCAR-8 cells, while it was slightly increased in the NCI/ADR-RES cells.

Western blots were used to confirm our findings with RT-PCR. While there were significant differences in transcript levels for *GPX4*, *NOX4*, *CHAC1*, *HMOX1,* and *NRF2* following ER treatment of OVCAR-8 or its resistant variant NCI/ADR-RES cells, we found no differences in their protein levels by Western blot (Figure 8C). This is somewhat surprising; however, our previous results with NO-generating drugs have shown that transcript levels do not always correlate with protein expression [42]. This may be due to rapid reactions of nitric oxide with proteins leading to post-translation modifications and degradation, as reported previously [43,44]. Our studies show that GPX4 protein expression was unchanged in these ovarian cells, suggesting that the enzyme activity was suppressed by ER and not its expression.

## 3. Discussion

Due to difficulties in curing clinical resistance and a lack of suitable combinational chemotherapy agents for ovarian cancers, ferroptosis-inducing agents are considered one of the novel approaches for successful therapy. This novel form of cell death is characterized by iron dependence and oxidative damage, resulting in cellular damage via lipid peroxidation, which results from decreased levels of cellular GSH due to the inhibition of the xCT-transporter by ER. Cystine is then converted to cysteine and ultimately to cellular GSH, which is utilized by GPX4 to reduce hydrogen peroxide, organic hydroperoxides, and lipid hydroperoxides. Thus, the inhibition of cystine import by ER decreases GSH synthesis, inhibiting GPX4 activity, producing lipid ROS, and inducing oxidative damage and ferroptosis. RSL3 and other related compounds/agents have been shown to directly inhibit GPX4, enhancing the process of ferroptosis.

We have shown that NCX4040, a non-steroidal nitric oxide donor, induces ferroptosis in human colon cancer cells, and this was significantly enhanced by both ER and RSL3 [42]. These observations then suggest that combinations of anticancer drugs with ferroptosis inducers may be effective for the treatment of both Ras-mutated and non-Ras-mutated tumor cells. Understanding the molecular mechanisms that regulate ferroptosis will help us design better combinations of drugs to treat human cancers. Towards this end, understanding the mechanisms of ER underlying the differential sensitivity of cell types to ER-induced ferroptosis is critical. Utilizing ER (and its recently synthesized analogs) as part of novel combinational strategies for cancer therapy requires elucidation of its mechanism of action in different cancer cell types.

In this study, we investigated and compared the sensitivity of human ovarian cancer OVCAR-8 and ADR-selected NCI/ADR-RES cells to ER. These ovarian tumors have a P121H Ras mutation [45,46] containing mutated p53 [47,48], and thus are markedly resistant to standard chemotherapy. These cell lines possess several advantages for elucidating the mechanisms of ER sensitivity. First, ADR-selected NCI/ADR-RES cells contain higher levels of various enzymes (e.g., SOD, catalase, glutathione-dependent peroxidase (GPX1), and transferase) that are involved in detoxifications of ROS. One mechanism of ER-induced cell death is suggested to depend on ROS formation, which induces lipid peroxidation and ferroptosis. If this scenario is correct, then NCI/ADR-RES cells would exhibit resistance to ER-induced ferroptosis and cell death. Additionally, NCI/ADR-RES cells overexpress P-gp. A recent article suggests that ER [23] and some of its analogs are substrates for P-gp [23,49], which would also contribute to the ability of NCI/ADR-RES cells to resist ER-dependent cell death.

In this study, we found that ER inhibits xCT-cystine/glutamate transporters in both cell lines at similar concentrations. We further observed that ER induces cell death in both cell lines, which was enhanced by RSL3 and attenuated by Ferrostatin-1, an inhibitor of ferroptosis. These observations strongly suggest that ER induces ferroptosis-mediated cell death in ovarian tumor cells. Intriguingly, ER is equally cytotoxic to both OVCAR-8 and its resistant variant NCI/ADR-RES cells, indicating that ER is not a substrate for p-gp. Our findings are similar to those recently reported by Frye et al. [49], but are in contrast to those reported by Zhou et al. [23], who utilized a Taxol-selected p-gp-overexpressing ovarian cell line which was significantly resistant to ER [23]. The reason for this discrepancy is unclear, but it may result from the difference in selecting agents (NCI/ADR-RES cells were selected using Adriamycin, not Taxol) and, thus, different mechanisms of resistance in addition to P-gP expression. 

While ER-induced ROS formation only in OVCAR-8 cells and ROS was not detected in the NCI/ADR-RES cells following treatment with ER at 4 h, NAC completely attenuated ER cytotoxicity in both cell types, indicating that ER cytotoxicity was ROS-mediated. Furthermore, both OVCAR-8 and NCI/ADR-RES generated similar amounts of lipid peroxides and MDA following treatment with RSL, a direct inhibitor of GPX4, indicating that GPX4 plays a vital role in the cytotoxicity of ER in these ovarian cells. This is further supported by our observations that RSL3 significantly enhanced the cytotoxicity of ER in both cell lines and was slightly more cytotoxic to NCI/ADR-RES cells. While ER-mediated cell death is clearly ROS-dependent in both cell types, additional pathways may also be operative to induce oxidative damage in NCI/ADR-RES cells, as ER was equally effective in inhibiting xCT-transporter in both cell types (Figure 4).

ER induces nitric oxide (^●^NO) formation in MDA-MB-231 breast cancer cells and has been implicated in the induction of ferroptosis [50]. It has been reported that ER-induced GSH depletion activated disulfide isomerase (PDI), which induced iNOS, resulting in an accumulation of cellular lipid ROS. Hou et al. have also shown that ER-induced GSH depletion leads to the activation of PDI, which then mediates ferroptosis by catalyzing nNOS dimerization, resulting in the accumulation of cellular ^●^NO, ROS, and lipid ROS, which ultimately causes ferroptotic cell death in immortalized HT22 mouse hippocampal neuronal cells [51]. We have also previously reported that the ^●^NO-generating drug, NCX4040, induces ferroptosis in colon cancer cells, which was further enhanced by ER [42]. These observations suggest that ER may induce/activate iNOS in NCI/ADR-RES cells to generate ^●^NO, which then reacts with O_2_^●−^ to generate NOOOH, which then participates in ER-induced ferroptotic cell death. These observations are further supported by our recent findings that ER reverses ADR resistance in NCI/ADR-RES cells without significantly modulating ADR cytotoxicity in OVCAR-8 cells [52]. Furthermore, the increase in ADR cytotoxicity in NCI/ADR-RES cells was inhibited by 1400 w, a specific inhibitor of iNOS, suggesting the formation of ^●^NO. We have shown that ^●^NO inhibits ATPase activity [53], including the ABC transporter [54,55]. Further, 1400 w had no effects on ADR cytotoxicity in OVCAR-8 cells, suggesting that ER may not induce/activate iNOS in OVCAR-8 cells; thus, the contributions of ^●^NO would be minimal in this cell line. However, this needs to be confirmed in future studies. Figure 9 summarizes our findings.

Our studies demonstrate that CHAC1 was similarly induced by ER in both the OVCAR-8 and NCI/ADR-RES cells. CHAC1 is known to hydrolyze cellular GSH, thereby depleting it [56]. This suggests that GSH levels are further reduced in NCI/ADR-RES cells, in addition to the inhibition of the xCT-transporter and GSR by ER. We did not observe GSH depletion at 4 h of treatment, and significant cell death was observed at 24 h with higher concentrations of ER (10, 20 µM). Our recent metabolomic analysis (ER at 2.5 µM) showed significant decreases in cellular GSH in both cell lines at 24 h (Kirkwood-Donelson et. al., manuscript in preparation).

In addition, our metabolomic studies indicated significant decreases in carnitines and carnitine derivatives in both cell lines. Carnitines transport long-chain fatty acids into mitochondria for beta oxidation and ATP production, and they protect cells from oxidative damage by safeguarding detoxifying enzymes (SOD, catalase, peroxidases) [57,58]. While speculative at this time, this may suggest that NCI/ADR-RES cells could become vulnerable to oxidative damage, as these cells have higher levels of detoxification enzymes. Collectively, our studies suggest that ER induces oxidative stress in both cell lines. However, ER-induced cellular oxidative damage may be mediated by additional pathways in NCI/ADR-RES cells, leading to ferroptotic cell death.

We also show that ER is not a substrate for p-gp and that it can be used to treat ovarian tumors harboring RAS mutations or showing MDR phenotypes in the clinic. Our studies further suggest that combinations of ER and RSL3 provide a promising approach for future ovarian cancer treatments. Further research into the mechanisms of ferroptosis will offer more therapeutic targets for clinical applications, potentially leading to the development of new drugs and therapeutic approaches.

Sorafenib, a kinase inhibitor, is currently used to treat unresectable liver carcinoma, advanced renal carcinoma, differentiated thyroid carcinoma, and ovarian cancers [59]. Sorafenib blocks xCT activity and GSH synthesis similarly to ER, inducing ferroptosis in solid tumors [60]. Cancer cells treated with cisplatin show decreases in GSH and inactivation of GPX4, initiating ferroptosis. The combination of cisplatin and ER has been shown to significantly improve antitumor efficiency. PRLX93936, an analogue of ER, has been tested in clinical trials. Co-treatment with cisplatin and PRLX93936 induces lipid peroxidation and Fe^2+^ production, promoting ferroptosis [61].

The clinical relevance of our work with erastin and RSL3 is significant, particularly in the context of cancer therapy. Ferroptosis inducers like erastin and RSL3 can be used to treat resistant tumors, as they employ different mechanisms of cell death compared to conventional chemotherapy, which often leads to chemotherapy resistance. Our studies indicate that the cell death induced by these agents is based on oxidative stress and iron, allowing for the selective targeting of tumors with higher iron levels.

Furthermore, erastin and similar inducers have been shown to sensitize various chemotherapeutics, including Adriamycin, Taxol, and cisplatin-based drugs. This suggests that erastin and RSL3 could be used in combination chemotherapy alongside radiotherapy or immune checkpoint inhibitors to enhance their effects. While beyond the scope of this study, combining erastin with other anticancer drugs, such as Adriamycin, could be valuable for treating ABC transporter-expressing tumors in clinical settings. We plan to initiate a study combining erastin with several ovarian tumor-active drugs, such as Taxol and platinum-based compounds, in various tumor models, including in vivo xenografts.

## 4. Materials and Methods

Adriamycin, the cystine analog, Seleno-L-cystine, the fluorescent molecule fluorescein O,O’-diacrylate, and the system xCT-inhibitor sulfasalazine were purchased from Sigma Aldrich (St. Louis, MO, USA). Erastin, (2-[1-[4-[2-(4-Chlorophenoxy)acetyl]-1-piperazinyl]ethyl]-3-(2-ethoxyphenyl)-4(3*H*)-quinazolinone), RSL3, Ferrostatin-1, and N-acetyl cysteine (NAC) were purchased from Cayman Chemicals (Ann Arbor, MI, USA) and were dissolved in DMSO. Stock solutions were stored at −80 °C. Fresh drug solutions, prepared from the stock solutions, were used in all experiments. Primary antibodies were obtained from Abcam (Boston, MA, USA).

### 4.1. Cell Culture

Authenticated human ovarian OVCAR-8 (WT) and ADR-selected OVCAR-8 cells (NCI/ADR-RES cells, OV-R) were obtained from the Division of Cancer Treatment and Diagnosis Tumor Repository, National Cancer Institute the NCI-Frederick Cancer Center (Frederick, MD, USA). Cells were grown in Phenol Red-free RPMI 1640 media (pH = 7.0) supplemented with 10% fetal bovine serum and antibiotics. Cell cultures were incubated in an incubator at 37 °C and 5% CO_2_ with saturating humidity. Cells were routinely used for 20–25 passages, after which the cells were discarded, and a new cell culture was started from the frozen stock.

### 4.2. Cytotoxicity Studies

The cytotoxicity studies were carried out with TiterGlo (Promega, Madison, WI, USA). For the TiterGlo assay, roughly 2500–3000 cells/well were seeded in opaque white 96-well plates and allowed to attach overnight. Cells were then treated with various concentrations of ER and incubated for 72 h. Following the 72 h incubations, the cytotoxicity of ER was determined, according to the manufacturer’s instructions.

In the Trypan Blue assay for cytotoxicity, about 25,000–50,000 cells/well were seeded onto a 6-well plate (in duplicate) and allowed to attach for 18 h. Various concentrations of drugs (ER or combinations of ER and minimally cytotoxic concentrations of Ferrostatin-1, (FeS), or RSL3, or NAC (100 µM) were added to the cells in fresh complete media (2 ML) and incubated for 24 or 48 h). When used, NAC, FeS, or RSL3 were preincubated with cells for 1–2 h before the addition of ER. DMSO (0.01–0.1%) was included as the vehicle control when used. Following trypsinization, surviving cells were collected and 15 µL of cell mixtures were combined with 15 µL of Trypan Blue and counted in a T20 automatic cell counter (Bio-Rad, Hercules, CA, USA).

### 4.3. Cystine Uptake Assay via xCT Antiporter

Further, 5000–75,000/well (OVCAR-8 and NCI-ADR/RES) cells were plated into a Greiner Fluotrac flat-bottomed 96-well black plate and allowed to attach for 18 h. The media was removed, and 200 µL of pre-warmed cystine analog solution (200 µM selenocysteine) was added to other wells, except to the controls. Various concentrations of ER (1–20 µM) and 250 and 500 µM sulfasalazine were added. To the controls, 200 µL of cystine-free, serum-free media was added. The cells were then incubated at 37 °C for 60 min. The supernatant was removed, and the cells were washed three times with 200 µL of ice-cold PBS. The supernatant was removed, and 60 µL of 100% methanol was added to each well. A working solution was prepared immediately before use, which consisted of 100 mM MES buffer (pH 6.0), 10 µM fluorescein O,O′-diacrylate, and 200 µM tris-(2-carboxyethyl)phosphine hydrochloride. Then, 200 µL of the working solution was added to each well, mixed by pipetting, and incubated for 30 min at 37 °C. Following incubation, fluorescence was read using a Bi-Tek Synergy microplate reader. Fluorescence intensity was measured at Ex/Em = 490/535 and samples were read in triplicate.

### 4.4. Flow Cytometric Analysis of Mitochondrial ROS

The formation of ROS was determined as described previously [22]. Briefly, the cells were loaded with MitoSox Red (5 uM final concentration; Life Technologies, Carlsbad, CA, USA) for 30 min at 37 °C, 5% CO_2_ atmosphere before the addition of ER. Cells were examined at 2 h intervals with the addition of Sytox Blue as a vital dye by flow cytometry. A LSRFortessa flow cytometer (Benton Dickinson, San Jose, CA, USA), equipped with FACSDiVa software version 1, was used to analyze all the samples. MitoSox and Sytox Blue were excited using a 561 nm and 405 nm laser and detected using a 610/20 nm and 450/50 nm filter, respectively. For each sample, 10,000 cells were analyzed using FACSDiVa software.

### 4.5. Lipid Peroxidation Assay

An assay for the peroxidation of cellular lipids was carried out by measuring the formation of malondialdehyde (MDA) using 2-thiobarbituric acid, as previously published [62,63]. Briefly, about 3–4.0 × 10^6^ cells were incubated with various concentrations of ER or ADR (10 µM) for 4 h at 37 °C. Following incubations, the reactions were stopped by adding 10% trichloroacetic acid (2 ML), and the mixtures were centrifuged (5 min at 1000 g). Aliquots (1.5 mL) of the supernatant fractions were then reacted with 1.5 ML of 2% 2-thiobarbituric acid, and the chromophore was developed at 90.0 °C for 10 min. After the samples were cooled, the absorbance at 532 nm was determined.

### 4.6. Flow Cytometric Analysis of Annexin-V Binding

Changes in membrane phosphatidylserine symmetry were determined using an Annexin-5 V binding assay kit (Trevigen, Gaithersburg, MD, USA), according to the manufacturer’s instructions. Briefly, cells were washed in 1X PBS, and then incubated with 1 µL Annexin-V FITC and propidium iodide (PI) in Annexin-V binding buffer for 15 min at room temperature. After this time, the samples were diluted with 1X binding buffer and examined immediately by flow cytometry. Cells were analyzed using an LSRFortessa flow cytometer (Benton Dickinson, San Jose, CA, USA) equipped with FACSDiVa software. Annexin-V FITC and PI were excited using a 488 nm and 561 nm laser and detected using a 530/30 nm and 582/15 nm filter, respectively. For each sample, 10,000 cells were analyzed using FACSDiVa software.

### 4.7. Real Time RT-PCR

The expression levels of the selected transcripts were examined by real-time polymerase chain reaction (RT-PCR) using absolute SYBR green ROX Mix (ThermoFisher Scientific, Rochester, NY, USA), as previously described [22]. Cells (OVCAR-8 or NCI/ADR-RES) were treated with minimally cytotoxic doses of ER (2.5 µM) for 0 h, 4 h, or 24 h. Total RNA was isolated using Trizol following treatment with ER and purified. Data were analyzed using the ΔΔCt method of relative quantification, in which cycle times were normalized to β-actin (or GADPH) from the same sample. Primers for the selected genes were purchased from Origene (Gaithersburg, MD, USA, Table 1). All real-time fluorescence detection was carried out on an iCycler (Bio-Rad, Hercules, CA, USA). Experiments were carried out three different times and the results are expressed as the mean ± SEM. Analyses were performed using an unpaired Student’s *t*-test and considered significant when *p* ≤ 0.05.

### 4.8. Western Blot Assay

Cell pellets following treatment with ER (2.5 µM) for 4 and 24 h were collected, washed (ice-cold PBS), and homogenized in 200 µL PierceTM RIPA (Thermo Fisher, Waltham, MA, USA 89900) supplemented with 25× Complete Protease Inhibitor, EDTA-free (Roche, Belmont, CA, USA, 11836170001). Samples were centrifuged for 15 min at 14,000 g at 4 °C, and the supernatant aliquoted and stored at −80 °C. The concentration of protein in the samples was determined using the PierceTM BCA Protein Assay (Thermo Fisher, 23225). Then, 15 µg protein was combined with NuPage LDS Sample Buffer (Invitrogen, Waltham, MA, USA, NP0007) and NuPage Sample Reducing Agent (Invitrogen, NP 0009) and incubated at 70 °C for 10 min. Samples were loaded onto a NuPAGE™ 4 to 12% Bis-Tris gel (Invitrogen, NP0336BOX) and run for 35 min at 200 V in 1× NuPage MES Run Buffer (Invitrogen, NP0002), to which NuPage Antioxidant (Invitrogen, NP0005) had been added. A Spectra™ Multicolor Broad Range Protein Ladder (Thermo Fisher) was used as a reference for protein size. Proteins were transferred from the gel to nitrocellulose membranes using the iBlot™ Gel Transfer Device (Thermo Fisher). Ponceau S solution (Sigma, St. Louis, MO, USA, P7170) was used to visualize total protein transfer. EveryBlot Blocking Reagent (Bio-Rad, Hercules, CA, USA, 120110020) was used to block membranes. Membranes were incubated with primary antibody overnight at 4 °C, diluted in EveryBlot Blocking Reagent: rabbit anti-vinculin (Novus, St. Louis, MO, USA, NBP2-20859; 1:1000), rabbit anti-GPX4 (abcam, Cambridge, UK; 1:2000), mouse anti-NRF2 (Santa Cruz Biotechnology, Dallas, TX, USA; 1:200), rabbit anti-Nox4 (abcam; 1:1000). Membranes were washed 3 × 5 min in 1× tris-buffered saline (TBS; Bio-Rad, 1706435) plus 0.1% tween-20 (Sigma, P7949) (TBST). Membranes were incubated for 45 min at room temperature with secondary antibody diluted in EveryBlot Blocking Reagent (Bio-Rad): goat-anti-rabbit HRP (Novus, NB7160; 1:5000) or goat-anti-mouse HRP (Invitrogen, 32230; 1:1000). Excess secondary antibody was removed by washing 3 × 5 min in TBS. HRP was visualized using SuperSignal™ West Pico Plus Chemiluminescent Substrate (Thermo Fisher, 34580).

### 4.9. Statistical Analysis

The results were expressed as mean ± SEM of a minimum of 3 independent experiments (n = 3). One-way analysis of variance (ANOVA) or Student’s *t*-test were used for statistical analysis using Graph Pad Prism, version 8 (GraphPad Software, Inc., La Jolla, CA, USA). For multiple comparisons, the Tukey’s multiple comparison’s test was utilized and considered statistically significant when *p* < 0.05.

## 5. Conclusions

In conclusion, our studies show that erastin (ER), a ferroptosis inducer, is not a substrate for P-gp and induces ferroptotic cell death in both OVCAR-8 and its P-gp-expressing NCI/ADR-RES tumor cells. While ROS were only detected in OVCAR-8 cells, significant oxidative damage was present in both cell types, as evidenced by the presence of several oxidative damage markers: *HMOX1/OX1*, *OGG1*, and *CHAC1*. Additionally, ER cytotoxicity was completely attenuated by NAC in both cell types, indicating a free radical-based mode of cell death. Our observations suggest that GPX4 may be central to ER-mediated cell death, as the direct inhibition of GPX4 by RSL3 enhanced ER-mediated lipid peroxidation in both cell types, significantly increasing ER-dependent cell death. These studies further indicate that ER and RSL3 are important modulators of ovarian cell death and may provide excellent combinatorial therapy for treating resistant cancers, including those with K-RAS mutations and/or expressing MDR phenotypes.

## Figures and Tables

**Figure 1 ijms-25-08666-f001:**
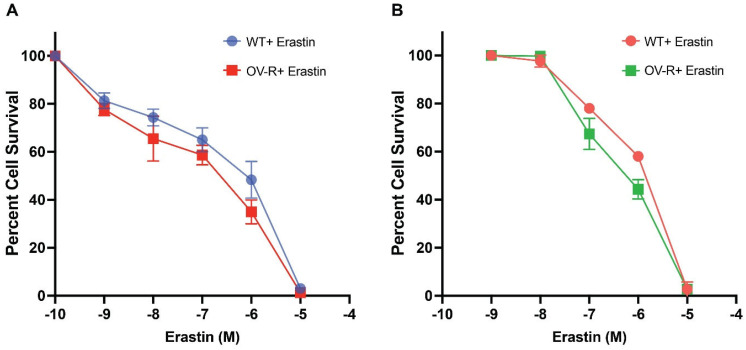
Cytotoxicity of erastin in OVCAR-8 and NCI/ADR-RES cells following 72 h of treatment using TiterGlo (**A**) and Trypan Blue (**B**) cytotoxicity assays.

**Figure 2 ijms-25-08666-f002:**
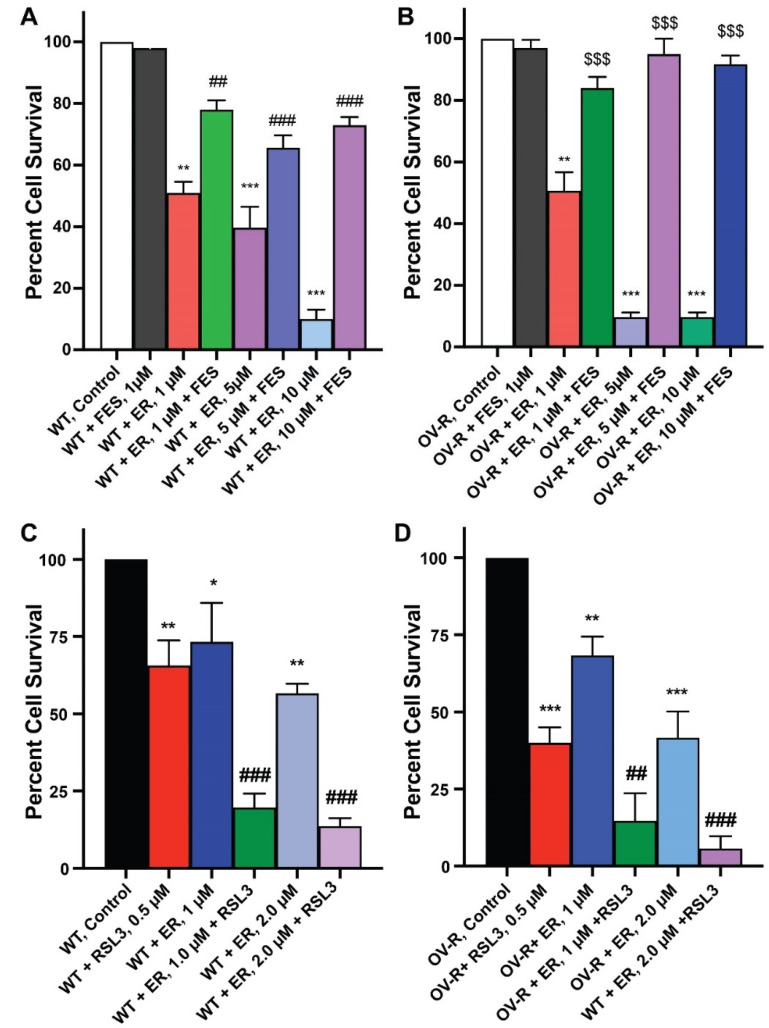
Effects of Ferrostatin-1 (**A**,**B**) on the cytotoxicity of erastin in ovarian cells following 48 h incubations. (**A**) OVCAR-8 and (**B**) NCI/ADR-RES cells, respectively. Effects of RSL3 (0.5 µM) on cytotoxicity of ER following 24 h of incubations (**C**) OVCAR-8 and (**D**) NCI/ADR-RES cells. *, **, and *** *p* values > 0.05, 0.005, and 0.001, respectively, compared to untreated control. ## and ### *p* values > 0.005 and 0.001, respectively, compared to treated FES or RSL3 alone to treated ER + FES or ER + RSL3. $$$ *p* values < 0.001, compared to treated ER alone.

**Figure 3 ijms-25-08666-f003:**
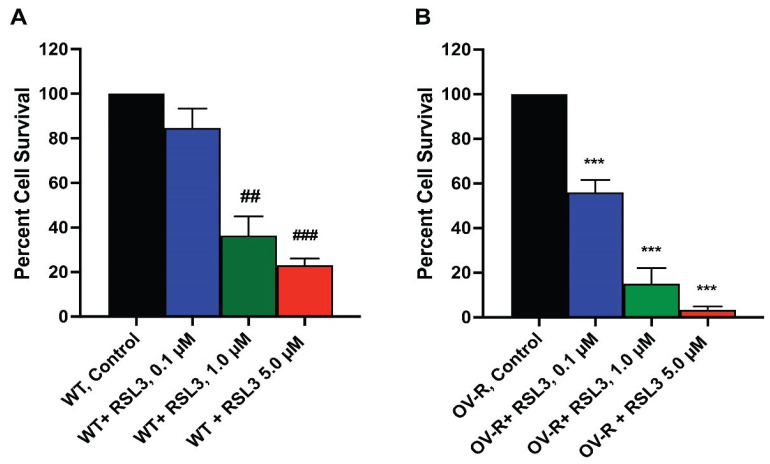
Dose dependence of RSL3 cytotoxicity in OVCAR-8 (**A**) and NCI/ADR-RES cells (**B**). The cells were incubated with different concentrations of RSL3 for 24 h. ## and ### <0.005 and 0.001, respectively, and *** *p* values < 0.001, compared to untreated control.

**Figure 4 ijms-25-08666-f004:**
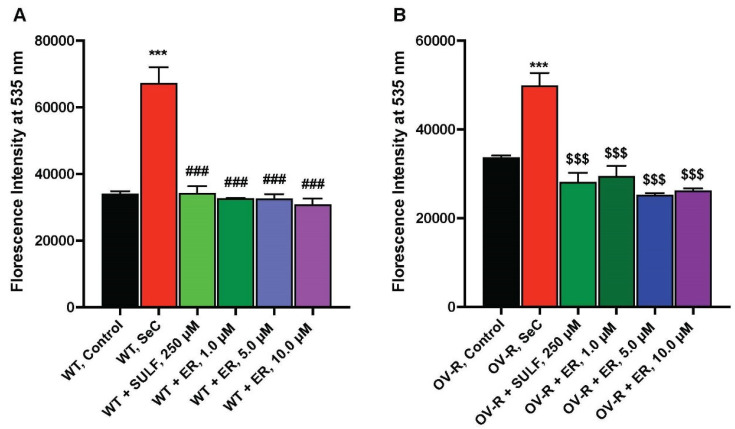
Effects of ER on the Xc-transporter in OVCAR-8 (**A**) and NCI/ADR-RES (**B**) cells. *** *p* values < 0.001, compared to untreated controls; ### and $$$ *p* values < 0.001 compared to Se-Cysteine treated cells.

**Figure 5 ijms-25-08666-f005:**
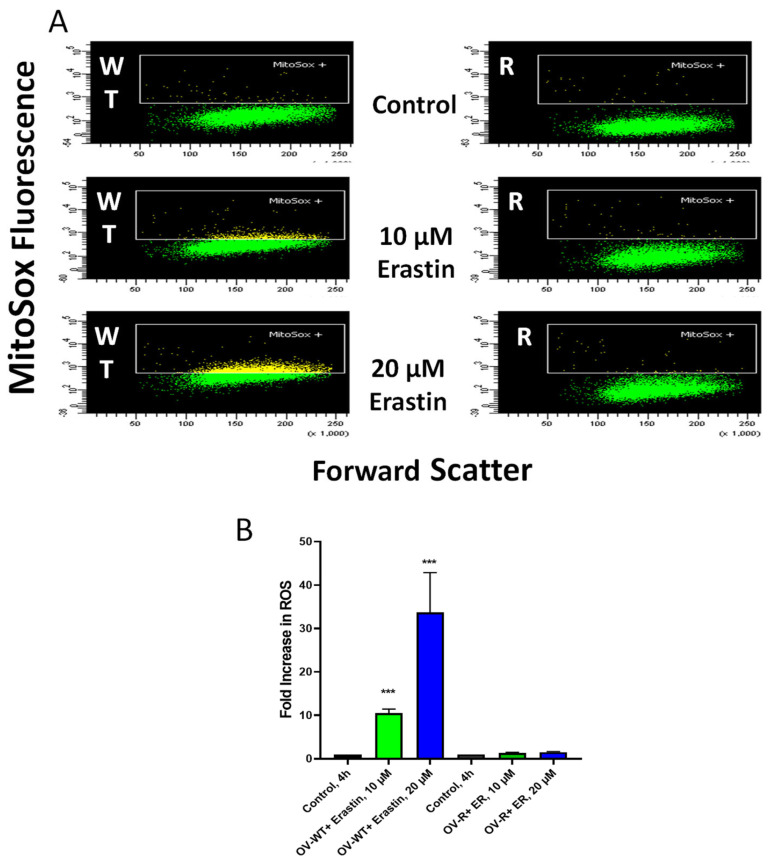
Formation of Mitosox+ cells in OVCAR-8 and NCI/ADR-RES cells following 4 h incubations with ER (**B**). A representative scatter plot (**A**) for OVCAR-8 and NCI/ADR-RES cells is shown here. *** *p* values < 0.001, respectively, compared to untreated control.

**Figure 6 ijms-25-08666-f006:**
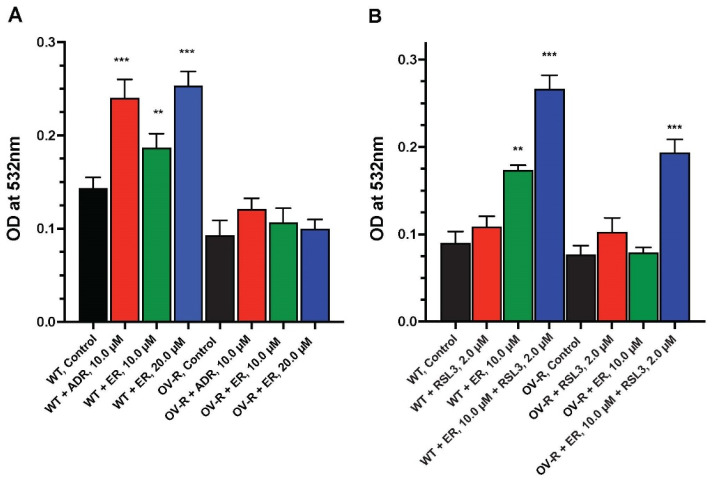
Dose dependence of ER-induced lipid peroxidation in OVCAR-8 and NCI/ADR-RES cells at 4 h (**A**) and effects of RSL3 on MDA formation (**B**). The MDA formation was measured at 532 mM. ** and *** *p* values < 0.005 and <0.001, respectively, compared to untreated control.

**Figure 7 ijms-25-08666-f007:**
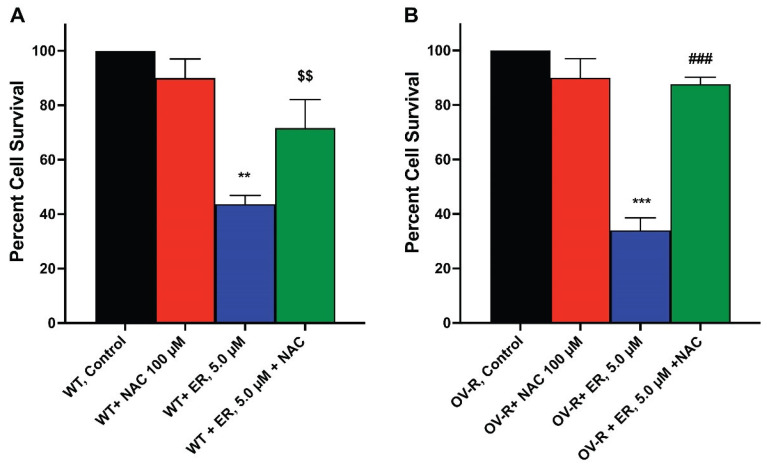
Effects of N-acetyl Cysteine (NAC) on ER cytotoxicity in OVCAR-8 (**A**) and NCI/ADR-RES cells (**B**). The cells were incubated with 100 µM NAC for 30 min before adding ER for 24 h. ** and *** *p* values < 0.005 and 0.001, compared to untreated control. $$ and ###, *p* values < 0.005 and 0.001, respectively, compared to ER values alone.

**Figure 8 ijms-25-08666-f008:**
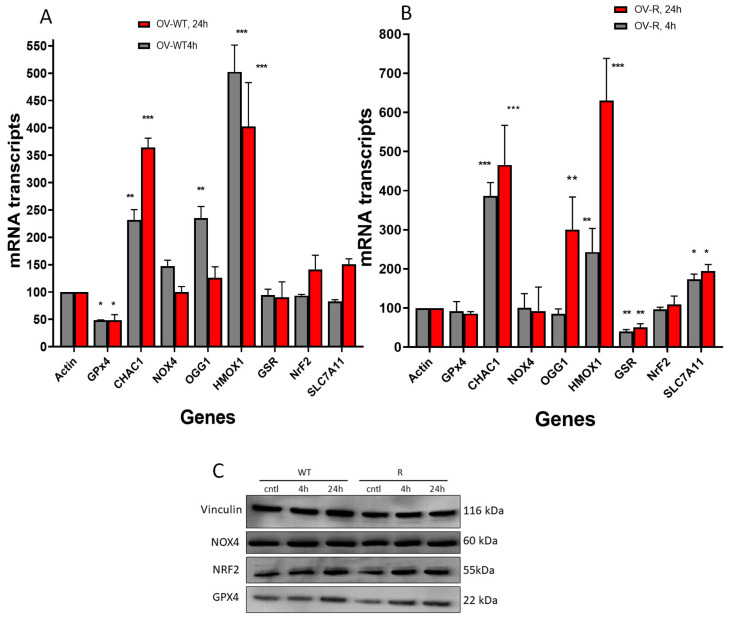
Effects of erastin (2.5 µM) on various oxidative and ferroptosis-related genes in OVCAR-8 (**A**) and NCI/ADR-RES cells (**B**) cells following treatment with erastin for 4 h and 24 h. Protein levels for GPX4, NrF2, and NOX4 following treatment with 2.5 µM for 4 and 24 h in OVCAR-8 and NCI/ADR-RES cells (**C**). * *p* < 0.05, ** and *** *p* values < 0.005 and 0.001, respectively, compared to control (β-Actin at 4 h and 24 h, respectively).

**Figure 9 ijms-25-08666-f009:**
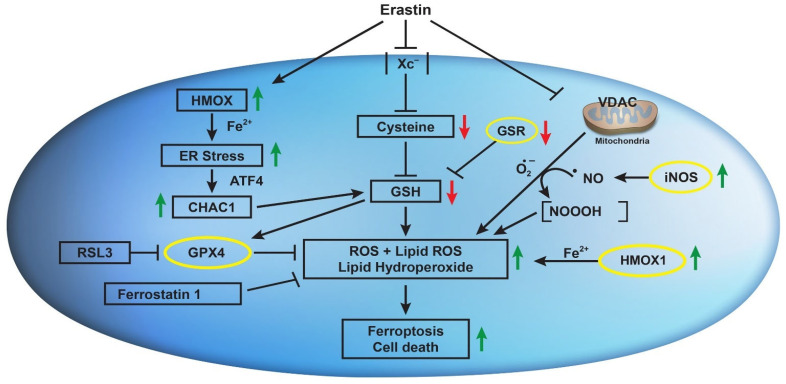
Effects of erastin on Xc- transporter, VDAC, *CHAC1*, *HMOX1*, iNOS, and their implications in erastin-induced lipid peroxidation and ferroptosis in OVCAR-8 and NCI/ADR-RES cells.

**Table 1 ijms-25-08666-t001:** Primers used and their sequences.

Beta-Actin	CACCATTGGCAATGAGCGGTTC-forward; AGGTCTTTGCGGATGTCCACGT-reverse
GPX4	ACAAGAACGGCTGCGTGGTGAA-forward; GCCACACACTTGTGGAGCTAGA-reverse
CHAC1	GTGGTGACGCTCCTTGAAGATC-forward; GAAGGTGACCTCCTTGGTATCG-reverse
NOX4	GCCAGAGTATCACTACCTCCAC-forward; CTCGGAGGTAAGCCAAGAGTGT-reverse
OGG1	GGCTCAACTGTATCACCACTGG-forward; GGCGATGTTGTTGTTGGAGGAAC-reverse
HMOX1	CCAGGCAGAGAATGCTGAGTTC-forward; AAGACTGGGCTCTCCTTGTTGC-reverse
GSR	TATGTGAGCCGCCTGAATGCCA-forward; CACTGACCTCTATTGTGGGCTTG-reverse
NRF2	CACATCCAGTCAGAAACCAGTGG-forward; GGAATGTCTGCGCCAAAAGCTG-reverse
SLC7A11	TCCTGCTTTGGCTCCATGAACG-forward; AGAGGAGTGTGCTTGCGGACAT-reverse
BCL2	ATCGCCCTGTGGATGACTGAGT-forward; GCCAGGAGAAATCAAACAGAGGC-reverse
BAX	TCAGGATGCGTCCACCAAGAAG-forward; TGTGTCCACGGCGGCAATCATC-reverse

## Data Availability

Data are contained within the article.

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
