# Peer review of "Mechanisms of Cell Death Induced by Erastin in Human Ovarian Tumor Cells"

_ijms, 2024, doi:10.3390/ijms25168666_

Round 1
Reviewer 1 Report
Comments and Suggestions for Authors
Hello colleagues, thank you for the opportunity to review this interesting manuscript on the mechanisms of erastin-induced cell death in ovarian tumor cells. As a senior medical researcher with years of experience, I'm happy to share my thoughts.
First off, I think the authors have done a nice job utilizing the OVCAR-8 and NCI/ADR-RES cell line models to really dissect the role of oxidative stress and ferroptosis in erastin's cytotoxic effects. Leveraging the antioxidant-rich NCI/ADR-RES line as a comparison was a clever approach that allows them to tease apart the importance of ROS generation in this process.
In terms of the experimental methods, I thought the combination treatments with ferroptosis modulators like ferrostatin-1 and RSL3 were particularly insightful. That kind of targeted mechanistic probing really helps strengthen the authors' conclusions about the centrality of ferroptosis in erastin's cell killing activity.
That said, I do think expanding the analysis to a broader panel of ovarian cancer cell lines, and even patient-derived samples, could help bolster the broader applicability of these findings. The current focus on just two specific cell lines, while enlightening, may limit the generalizability of the conclusions.
Additionally, I would encourage the authors to dig deeper into the transcriptional and post-translational regulation of key ferroptosis pathway components. A more comprehensive assessment of how erastin modulates the expression and activities of players like GPX4, SLC7A11, and others could further elucidate the underlying mechanisms.
Finally, I think the authors could do a bit more to contextualize the clinical relevance of this work. Highlighting the potential for erastin, or combination therapies with ferroptosis inducers, to address drug-resistant ovarian cancers would help strengthen the translational implications.
Overall, though, I commend the authors on a well-designed and thoughtfully executed study. The insights gained here represent an important step forward in understanding how we might leverage ferroptosis pathways for more effective ovarian cancer treatment. I wish the team the best of luck as they continue to build on this foundation.
Please let me know if you have any other questions!
Comments on the Quality of English LanguageHello colleagues, thank you for the opportunity to review this interesting manuscript on the mechanisms of erastin-induced cell death in ovarian tumor cells. As a senior medical researcher with years of experience, I'm happy to share my thoughts.
First off, I think the authors have done a nice job utilizing the OVCAR-8 and NCI/ADR-RES cell line models to really dissect the role of oxidative stress and ferroptosis in erastin's cytotoxic effects. Leveraging the antioxidant-rich NCI/ADR-RES line as a comparison was a clever approach that allows them to tease apart the importance of ROS generation in this process.
In terms of the experimental methods, I thought the combination treatments with ferroptosis modulators like ferrostatin-1 and RSL3 were particularly insightful. That kind of targeted mechanistic probing really helps strengthen the authors' conclusions about the centrality of ferroptosis in erastin's cell killing activity.
That said, I do think expanding the analysis to a broader panel of ovarian cancer cell lines, and even patient-derived samples, could help bolster the broader applicability of these findings. The current focus on just two specific cell lines, while enlightening, may limit the generalizability of the conclusions.
Additionally, I would encourage the authors to dig deeper into the transcriptional and post-translational regulation of key ferroptosis pathway components. A more comprehensive assessment of how erastin modulates the expression and activities of players like GPX4, SLC7A11, and others could further elucidate the underlying mechanisms.
Finally, I think the authors could do a bit more to contextualize the clinical relevance of this work. Highlighting the potential for erastin, or combination therapies with ferroptosis inducers, to address drug-resistant ovarian cancers would help strengthen the translational implications.
Overall, though, I commend the authors on a well-designed and thoughtfully executed study. The insights gained here represent an important step forward in understanding how we might leverage ferroptosis pathways for more effective ovarian cancer treatment. I wish the team the best of luck as they continue to build on this foundation.
Please let me know if you have any other questions!
Author Response
Reviewer-1
First off, I think the authors have done a nice job utilizing the OVCAR-8 and NCI/ADR-RES cell line models to really dissect the role of oxidative stress and ferroptosis in erastin's cytotoxic effects. Leveraging the antioxidant-rich NCI/ADR-RES line as a comparison was a clever approach that allows them to tease apart the importance of ROS generation in this process.
Thank you for your kind words.
In terms of the experimental methods, I thought the combination treatments with ferroptosis modulators like ferrostatin-1 and RSL3 were particularly insightful. That kind of targeted mechanistic probing really helps strengthen the authors' conclusions about the centrality of ferroptosis in erastin's cell killing activity.
Again, Thank you for your kind words, appreciated very much.
That said, I do think expanding the analysis to a broader panel of ovarian cancer cell lines, and even patient-derived samples, could help bolster the broader applicability of these findings. The current focus on just two specific cell lines, while enlightening, may limit the generalizability of the conclusions.
We had already planned this and have ordered various ovarian cancer cell lines from ATCC, including SW626 (a Ras mutant), CRL-1572 and Caov-3, which are part of ovarian cancer cell panel for this purpose.
Additionally, I would encourage the authors to dig deeper into the transcriptional and post-translational regulation of key ferroptosis pathway components. A more comprehensive assessment of how erastin modulates the expression and activities of players like GPX4, SLC7A11, and others could further elucidate the underlying mechanisms.
In our experiments, SLC7A11 was not significantly modulated by Erastin in OVCAR-8 and NCI/ADR-RES cells. This does not imply that this gene is unaffected by Erastin or other ferroptosis inducers in other ovarian cell lines. We plan to examine this in the new cell lines we have ordered. Additionally, we have an ongoing gene expression profiling study in colon cancer cell lines.
Finally, I think the authors could do a bit more to contextualize the clinical relevance of this work. Highlighting the potential for erastin, or combination therapies with ferroptosis inducers, to address drug-resistant ovarian cancers would help strengthen the translational implications.
This was, in part, was included in our manuscript in the discussion section. However, we have now expanded this more in the revised manuscript, page-25-26, last paragraph.
Overall, though, I commend the authors on a well-designed and thoughtfully executed study. The insights gained here represent an important step forward in understanding how we might leverage ferroptosis pathways for more effective ovarian cancer treatment. I wish the team the best of luck as they continue to build on this foundation.
Even after 50 years of publishing and working in science, it is always nice to hear nice comments, thank you.
Please let me know if you have any other questions!
Reviewer 2 Report
Comments and Suggestions for Authors
The aim of the study by Sinha et al. is to demonstrate the cytotoxicity of erastin against two cell lines, i.e. mutant human ovarian tumor OVCAR-8 and NCI/ADR-RES, P-glycoprotein-expressing cells. The authors proposed an explanation of the mechanisms of cell death pointing to the important role of glutathione peroxidase 4 (GPX4). The strength of the article is the large number of analyses. The paper needs minor revisions before acceptance.
Minor comments:
Line 6 - In the affiliation, please indicate the country of origin of the authors.
Line 72 - The abbreviation LOOH appears only once in the text. Therefore, I do not see the point of introducing it.
Line 97 - did the authors mean glutathione peroxidase 1?
Line 107 and the rest of MM - with reagents, as a rule, the country of origin should be given.
Line 192 - can the authors say more about how the primers were designed and validated?
Line 213- please provide information on how the specificity of the primary antibodies used was checked?
Line 223-The description of statistical methods is quite laconic. What test was used to check normal distribution? How was homogeneity of variance checked? What post-hoc test was used?
Supplementary figures A and B - please attach full un-cropped scans of blots.
Author Response
The aim of the study by Sinha et al. is to demonstrate the cytotoxicity of erastin against two cell lines, i.e. mutant human ovarian tumor OVCAR-8 and NCI/ADR-RES, P-glycoprotein-expressing cells. The authors proposed an explanation of the mechanisms of cell death pointing to the important role of glutathione peroxidase 4 (GPX4). The strength of the article is the large number of analyses. The paper needs minor revisions before acceptance.
Minor comments:
Line 6 - In the affiliation, please indicate the country of origin of the authors.
Done
Line 72 - The abbreviation LOOH appears only once in the text. Therefore, I do not see the point of introducing it.
Has been corrected.
Line 97 - did the authors mean glutathione peroxidase 1?
Yes, Glutathione Peroxidase1 and is now corrected.
Line 107 and the rest of MM - with reagents, as a rule, the country of origin should be given.
This has been now included.
Line 192 - can the authors say more about how the primers were designed and validated?
These were purchased from Origene, Gaithersburg, MD, USA. We used these primers as they were also used by other investigators (and published and cited) previously as indicated by the manufacture.
Line 213- please provide information on how the specificity of the primary antibodies used was checked?
Again, these antibodies have been used (and published and cited) by other investigators according to the manufacture.
Line 223-The description of statistical methods is quite laconic. What test was used to check normal distribution? How was homogeneity of variance checked? What post-hoc test was used?
As described in the manuscript, tests for normal distribution were conducted using both Student’s t-test and one-way ANOVA. A p-value of less than 0.05 typically indicates a violation of the assumption of normality. The data presented in this manuscript were derived from more than three independent experiments to determine p-values. Most of the analyses in these experiments were performed using computer programs with built-in statistical analysis tools.
Supplementary figures A and B - please attach full un-cropped scans of blots.
We have included uncropped total protein blots for figures A, B, and C. This is not a supplementary figure; it is the journal (MDPI) policy to submit uncropped original figures as proof of western blots. We have previously submitted similar blots to many journals, including Cancers and Cells, without any issues. I had some problems loading these gels and asked our contact editor, Ms. Nisareefah Benyakart, to upload them. She must have mistakenly included them as a supplementary figure.
Round 2
Reviewer 1 Report
Comments and Suggestions for Authors
This study aimed to investigate the mechanisms of cell death induced by Erastin (ER) in human ovarian tumor cells, specifically focusing on the role of ferroptosis in both OVCAR-8 and NCI/ADR-RES cell lines. The study highlights the potential of ER as a therapeutic agent for ovarian cancers, especially those resistant to conventional treatments.
## Experimental Methods
The study employed a series of experiments to achieve its objectives:
* **Cytotoxicity studies:** TiterGlo and Trypan Blue assays were used to assess ER's cytotoxicity in both cell lines.
* **Cystine uptake assay:** This assay measured the effect of ER on cystine uptake via the xCT antiporter, a crucial component of ferroptosis.
* **Flow cytometric analysis of mitochondrial ROS:** MitoSox Red staining was used to quantify mitochondrial ROS production, a key indicator of ferroptosis.
* **Lipid peroxidation assay:** This assay measured the formation of malondialdehyde (MDA), a marker of lipid peroxidation, another hallmark of ferroptosis.
* **Flow cytometric analysis of Annexin-V binding:** This assay assessed the changes in membrane phosphatidylserine symmetry, indicating apoptosis.
* **Real-time RT-PCR:** This technique measured the expression levels of genes associated with ferroptosis and oxidative stress.
* **Western blot assay:** This technique evaluated the protein expression levels of key ferroptosis-related proteins.
## Limitations
* The study did not investigate the potential off-target effects of ER or other drugs used.
* The study did not explore the in vivo efficacy of ER in animal models of ovarian cancer.
* The study did not investigate the potential mechanisms of resistance to ER-induced ferroptosis.
## Suggestions for Improvement
1. **Investigate the off-target effects of ER and other drugs used.** This would provide a more comprehensive understanding of the potential side effects of these agents.
2. **Evaluate the in vivo efficacy of ER in animal models of ovarian cancer.** This would provide valuable information about the potential clinical translation of ER-based therapy.
3. **Explore the potential mechanisms of resistance to ER-induced ferroptosis.** This would help identify strategies to overcome resistance and improve the effectiveness of ER-based therapy.
1. "Erastin (ER) induces cell death through formation of reactive oxygen species (ROS) and subsequent ferroptosis." - **Change "subsequent" to "resulting" for better clarity.**
2. "We used K-Ras mutant human ovarian tumor OVCAR-8 and NCI/ADR-RES, P-glycoprotein-expressing cells, to study the mecha￾nisms of ER-induced cell death." - **Change "mecha￾nisms" to "mechanisms" for correct spelling.**
3. "We found that ER was equally cytotoxic to both cells." - **Change "equally" to "similarly" for better accuracy, as the cytotoxicity values were not identical.**
4. "Ferrostatin, an inhibitor of ferroptosis, attenuated ER cytotoxicity." - **Change "attenuated" to "reduced" for clearer communication.**
5. "RSL3, an inducer of ferroptosis, significantly enhanced ER cytotoxicity in both cells." - **Change "significantly" to "markedly" for better emphasis.**
* The study employed a comprehensive set of experiments to investigate the mechanisms of ER-induced cell death.
* The study used two different ovarian cancer cell lines, providing insights into the potential broader applicability of ER-based therapy.
* The study provided valuable information about the role of ferroptosis in ER-induced cell death.
* The study did not investigate the potential off-target effects of ER or other drugs used.
* The study did not explore the in vivo efficacy of ER in animal models of ovarian cancer.
* The study did not investigate the potential mechanisms of resistance to ER-induced ferroptosis.
The authors should be commended for their thorough investigation into the mechanisms of ER-induced cell death in ovarian cancer cells. Their findings provide valuable insights into the potential of ER as a novel therapeutic agent for this deadly disease. Further research is warranted to address the limitations of this study and to translate these findings into clinical applications.
Comments on the Quality of English LanguageThis study aimed to investigate the mechanisms of cell death induced by Erastin (ER) in human ovarian tumor cells, specifically focusing on the role of ferroptosis in both OVCAR-8 and NCI/ADR-RES cell lines. The study highlights the potential of ER as a therapeutic agent for ovarian cancers, especially those resistant to conventional treatments.
## Experimental Methods
The study employed a series of experiments to achieve its objectives:
* **Cytotoxicity studies:** TiterGlo and Trypan Blue assays were used to assess ER's cytotoxicity in both cell lines.
* **Cystine uptake assay:** This assay measured the effect of ER on cystine uptake via the xCT antiporter, a crucial component of ferroptosis.
* **Flow cytometric analysis of mitochondrial ROS:** MitoSox Red staining was used to quantify mitochondrial ROS production, a key indicator of ferroptosis.
* **Lipid peroxidation assay:** This assay measured the formation of malondialdehyde (MDA), a marker of lipid peroxidation, another hallmark of ferroptosis.
* **Flow cytometric analysis of Annexin-V binding:** This assay assessed the changes in membrane phosphatidylserine symmetry, indicating apoptosis.
* **Real-time RT-PCR:** This technique measured the expression levels of genes associated with ferroptosis and oxidative stress.
* **Western blot assay:** This technique evaluated the protein expression levels of key ferroptosis-related proteins.
## Limitations
* The study did not investigate the potential off-target effects of ER or other drugs used.
* The study did not explore the in vivo efficacy of ER in animal models of ovarian cancer.
* The study did not investigate the potential mechanisms of resistance to ER-induced ferroptosis.
## Suggestions for Improvement
1. **Investigate the off-target effects of ER and other drugs used.** This would provide a more comprehensive understanding of the potential side effects of these agents.
2. **Evaluate the in vivo efficacy of ER in animal models of ovarian cancer.** This would provide valuable information about the potential clinical translation of ER-based therapy.
3. **Explore the potential mechanisms of resistance to ER-induced ferroptosis.** This would help identify strategies to overcome resistance and improve the effectiveness of ER-based therapy.
1. "Erastin (ER) induces cell death through formation of reactive oxygen species (ROS) and subsequent ferroptosis." - **Change "subsequent" to "resulting" for better clarity.**
2. "We used K-Ras mutant human ovarian tumor OVCAR-8 and NCI/ADR-RES, P-glycoprotein-expressing cells, to study the mecha￾nisms of ER-induced cell death." - **Change "mecha￾nisms" to "mechanisms" for correct spelling.**
3. "We found that ER was equally cytotoxic to both cells." - **Change "equally" to "similarly" for better accuracy, as the cytotoxicity values were not identical.**
4. "Ferrostatin, an inhibitor of ferroptosis, attenuated ER cytotoxicity." - **Change "attenuated" to "reduced" for clearer communication.**
5. "RSL3, an inducer of ferroptosis, significantly enhanced ER cytotoxicity in both cells." - **Change "significantly" to "markedly" for better emphasis.**
* The study employed a comprehensive set of experiments to investigate the mechanisms of ER-induced cell death.
* The study used two different ovarian cancer cell lines, providing insights into the potential broader applicability of ER-based therapy.
* The study provided valuable information about the role of ferroptosis in ER-induced cell death.
* The study did not investigate the potential off-target effects of ER or other drugs used.
* The study did not explore the in vivo efficacy of ER in animal models of ovarian cancer.
* The study did not investigate the potential mechanisms of resistance to ER-induced ferroptosis.
The authors should be commended for their thorough investigation into the mechanisms of ER-induced cell death in ovarian cancer cells. Their findings provide valuable insights into the potential of ER as a novel therapeutic agent for this deadly disease. Further research is warranted to address the limitations of this study and to translate these findings into clinical applications.
Author Response
Reviewer-3
Investigate the off-target effects of ER and other drugs used.** This would provide a more comprehensive understanding of the potential side effects of these agents.
Thank you for your kind and encouraging words. This topic is beyond the scope of this manuscript, but we will aim to address it in future research. Specifically, we plan to investigate this during our in vivo studies, combining Erastin with other anticancer drugs in an ovarian tumor model.
Evaluate the in vivo efficacy of ER in animal models of ovarian cancer.** This would provide valuable information about the potential clinical translation of ER-based therapy.
This is already planned.
**Explore the potential mechanisms of resistance to ER-induced ferroptosis.** This would help identify strategies to overcome resistance and improve the effectiveness of ER-based therapy.
In the model system used here (OVCAR-8 and NCI/ADR-RES cells), no significant resistance was observed. Erastin is also not a substrate for P-170 glycoprotein. However, we did find some de novo resistance to Erastin in colon cancer cells, and that work is currently ongoing.
Certain English words have been changed/corrected as suggested.